# Potential for Duplexed, In-Tandem gRNA-Mediated Suppression of Two Essential Genes of Tomato Leaf Curl New Delhi Virus in Crop Plants

**DOI:** 10.3390/pathogens14070679

**Published:** 2025-07-10

**Authors:** Saher Naveed, Judith K. Brown, Muhammad Mubin, Nazir Javed, Muhammad Shah Nawaz-ul-Rehman

**Affiliations:** 1Virology Lab, Center of Agricultural Biochemistry and Biotechnology, University of Agriculture, Faisalabad 38000, Pakistan; sahernaveed9@gmail.com (S.N.); mmubin@uaf.edu.pk (M.M.); 2School of Plant Sciences, The University of Arizona, Tucson, AZ 85721, USA; jbrown@ag.arizona.edu; 3Department of Plant Pathology, University of Agriculture, Faisalabad 38000, Pakistan

**Keywords:** begomovirus, CRISPR-Cas9, duplex guide RNA, single nucleotide polymorphism, ToLCNDV

## Abstract

Tomato leaf curl New Delhi virus (ToLCNDV) is among the most prevalent and widely distributed begomovirus infecting chili pepper (*Capsicum annuum*) and tomato in the Indian subcontinent. In this study, a guide RNA (gRNA) sequence-CRISPR-Cas9 approach was used to target and cleave two essential coding regions in the begomovirus genome. The gRNAs were designed to target conserved regions of the ToLCNDV replication-associated protein (*rep*) *gene* or ORF AC1, and/or the coat protein (*cp*) gene or AV1 ORF, respectively. Based on an alignment of 346 representative ToLCNDV genome sequences, all predicted single nucleotide polymorphisms off-target sites were identified and eliminated as potential gRNA targets. Based on the remaining genome regions, four candidate gRNAs were designed and used to build gRNA-Cas9 duplexed constructs, e.g., containing two gRNAs cloned in tandem, in different combinations (1–4). Two contained two gRNAs that targeted the coat protein gene (*cp*; *AV1 ORF*), while the other two constructs targeted both the *cp* and replication-associated protein gene (*rep*; *AC1 ORF*). These constructs were evaluated for the potential to suppress ToLCNDV infection in *Nicotiana benthamiana* plants in a transient expression-transfection assay. Among the plants inoculated with the duplexed gRNA construct designed to cleave ToLCNDV-AV1 or AC1-specific nucleotides, the construct designed to target both the *cp* (293–993 nt) and *rep* (1561–2324) showed the greatest reduction in virus accumulation, based on real-time quantitative PCR amplification, and attenuated disease symptoms, compared to plants inoculated with the DNA-A component alone or mock-inoculated, e.g., with buffer. The results demonstrate the potential for gRNA-mediated suppression of ToLCNDV infection in plants by targeting at least two viral coding regions, underscoring the great potential of CRISPR-Cas-mediated abatement of begomovirus infection in numerous crop species.

## 1. Introduction

Single-stranded DNA viruses belonging to the genus *Begomovirus* (family *Geminiviridae*) are among the most damaging plant viruses to vegetable, fiber, and root crops in temperate, subtropical, and tropical locales throughout the world. They represent the largest genus within the *Geminiviridae*, comprising >500 virus species [1]. Begomoviruses infect dicotyledonous plants. In the New World, viruses having a bipartite genome consisting of a DNA-A and DNA-B component are endemic, while bipartite begomoviruses and those having a monopartite genome consisting of a DNA-A component are endemic in the Old World [2].

In the Indian subcontinent, the tomato leaf curl disease (ToLCD) is caused by a highly damaging, recently emergent bipartite begomovirus pathogen that infects tomato and chili crops, and many other hosts, causing nearly complete yield loss in infected plants, especially when infection occurs during early growth stages [3,4]. The ToLCD was first reported in tomato crops in India in 1948 [5]. Since then, leaf curl disease has been documented in solanaceous crops in Asia, Europe, North Africa, the Pacific Rim, and Sub-Saharan Africa, where it has been associated with several monopartite begomoviruses, and occurs in single or mixed infections [6,7,8]. At least thirteen begomovirus species have been associated with ToLCD in the Indian subcontinent [2], with the bipartite tomato leaf curl New Delhi virus (ToLCNDV) being one of the most prevalent and widespread. Begomoviruses are transmitted in a persistent, circulative manner by the whitefly vector *Bemisia tabaci* (Genn.) cryptic species complex [9].

The first ToLCNDV genome sequence was determined in 1995 from symptomatic tomato plants in India [10], and since then, the virus has been identified in nearly all solanaceous crops and cucurbits [11,12,13,14]. In Pakistan, ToLCNDV has been associated with stunted plant growth, upward leaf curling, and vein thickening [15] of cultivated tomato, chili, and cucumber plants [16,17], and at least forty species are hosts of ToLCNDV [18]. Infection by ToLCNDV can result in a 60–100% reduced yield, depending on the crop [19]. From Asia, the virus spread to the Mediterranean region, including Greece, Italy, Morocco, Spain [20], Italy [21], Morocco [22], Thailand, and Tunisia [23], where it causes yield loss in zucchini and other cucurbitaceous species. This has led to the recognition of ToLCNDV as a quarantine pest by the European Plant Protection Organization (EPPO) [12]. Other studies have reported the association of the chilli leaf curl betasatellite (ChiLCB) with ToLCNDV [15]. Symptoms are more severe in plants infected with both ToLCNDV and ChLCB, compared to ToLCNDV alone [24]. Also, pepper leaf curl Lahore virus (PepLCLaV) has been detected in symptomatic chili plants, as well as the ToLCNDV DNA-B component [25], indicating that symptoms in chili peppers can be associated with more than one begomovirus. Thus, the development of ToLCNDV-resistant cucurbitaceous and solanaceous cultivars has become essential for food security.

Advancements in the precision of genome editing technology that deploys clustered regularly interspaced short palindromic repeat (CRISPR) and protein 9 (Cas9) for genome editing that can produce sequence-specific double-stranded DNA breaks [26]. The CRISPR-Cas system type II, adopted from *Streptococcus pyogenes,* is among the most frequently used systems for inducing double-stranded (ds) DNA breaks in a gene of interest [27]. CRISPR-Cas systems have been used to alter genes in *Arabidopsis thaliana*, rice, sorghum, tobacco, tomato, and wheat plants for the development of virus resistance as well as trait-associated genome alterations [28,29,30,31,32]. In several study systems, such directed mutations have been shown to be heritable in the next generation, thereby producing stably-modified genomes in plants such as *A. thaliana,* rice, and tomato [33,34].

Initial reports of CRISPR/Cas9 targeting the *Beet severe curly top virus* (BSCTV) in model plants demonstrated a high level of resistance to viral infection. In this study, the effectiveness of viral resistance varied depending on the targeted genome sites [35]. Further experiments using single guide RNAs (sgRNAs) targeting individual open reading frames of viruses such as *Wheat dwarf virus*, *Tomato yellow leaf curl virus*, *Cotton leaf curl Kokhran virus*, and *Cotton leaf curl Multan virus* produced varying levels of resistance, ranging from approximately 25% to 80% [36]. However, in cassava plants, Mehta et al. (2019) found that CRISPR/Cas9 constructs based on sgRNAs failed to confer resistance to cassava mosaic viruses [37]. Of the six different sgRNA constructs tested that targeted both virion and complementary strand genes, none provided effective protection, leading to the hypothesis that the replication mechanism of geminiviruses may override the mutations introduced by a single gRNA CRISPR/Cas9-mediated modification, thereby reducing the effectiveness of the strategy [38]. The results confirmed that using duplexed gRNAs to target viral genomes could prevent the emergence of escape mutants, in contrast to the single gRNA approach [39]. Given that single gRNAs have failed to confer durable resistance, while the use of duplexed gRNAs has achieved apparent full protection, it is of interest to explore a duplexed gRNA approach to suppress infection of ToLCNDV, a begomovirus that has a broad host range, is particularly widespread and highly damaging to many crop species.

In ToLCNDV, the AV1 gene is involved in encapsidation of the viral genome and is essential for cell-to-cell movement in plant hosts, while AC1 is essential for initiation of viral replication. The objective of this study was to design a CRISPR-Cas9 cassette expressing guide RNAs that specifically modify the ToLCNDV AV1 and AC1 genes, thereby interfering with virus replication and with cell-to-cell movement and encapsidation, respectively.

Initially, the gRNA targets selected from AV1 (*n* = 8) and AC1 (*n* = 5) were tested in planta for the ability to specifically cleave the ToLCNDV genome. Based on the experimental results, the most promising gRNAs were used to construct a duplexed gRNA-Cas9 cassette containing two gRNAs, each homologous to the ToLCNDV AV1 or AC1 sequence, respectively. The goal was to interfere with ToLCNDV infection of the host plant by reducing virus accumulation, and cell-to-cell and systemic spread, and ultimately, lead to attenuated or no disease symptoms. In this study, *N. benthamiana* plants were used as a surrogate virus-susceptible host for evaluating the most promising gRNA sequences for future deployment through an analogous approach to confer resistance in cultivated ToLCNDV hosts, such as tomato and chili pepper.

## 2. Materials and Methods

### 2.1. Genome Cloning and Construction of Tomato Leaf Curl New Delhi Virus

Previously, the ToLCNDV DNA-A and DNA-B components have been cloned and sequenced from infected chili pepper plants in Pakistan that exhibited begomovirus-like symptoms [40]. The ToLCNDV DNA-A and DNA-B (GenBank accession no. DQ116883) components were cloned into the pTZ57/R plasmid vector at *Bam*HI and *Pst*I restriction sites, respectively (Appendix A). Partial clones of ToLCNDV DNA-A and DNA-B components, respectively, were constructed in the pGreen0029 plasmid vector [41] (Appendix A). In the first step, a full-length clone of ToLCNDV DNA-A (~2.8 kb) was digested using *Bam*HI and *Kpn*I to yield two fragments of ~1.6 kbp (harboring origin of replication) and 1.2 Kbp in size. The 1.6 Kbp fragment was ligated into the *Bam*HI and *Kpn*I digested pGreen0029 plasmid vector to yield the partial clone, A. Subsequently, the full-length DNA-A from pTZ57/R was excised with *Kpn*I and ligated to the cloned fragment A at the *Kpn*I site, producing an infectious, in-tandem 1.75-mer (1.6 kbp + full-length 2.8 kbp), designated pMVL1113.

Similarly, a partial dimer of the DNA-B component was constructed using the *Pst*I and *Kpn*I sites, respectively. Briefly, the full-length clone (~2.7 kbp) was digested with *Pst*I and *Kpn*I, yielding a 2.2 (containing the origin of replication, or ‘ori’) and a 443 bp fragment, respectively. The 2.2 kbp fragment was ligated into the *Pst*I-*Kpn*I digested pGreen0029 plasmid vector to create a ‘partial B component’. The full-length DNA-B component, cloned into the pTZ57/R plasmid vector, was excised by restriction digestion with *Pst*I and ligated to the ‘partial B component’ at the corresponding *Pst*I site, yielding a partial dimer of 2.2 kbp + full-length 2.7 kbp in size, designated pMVL1116. Cells of *Agrobacterium tumefaciens* strain GV3103 were transformed with the ToLCNDV DNA-A component, the DNA-B component partial dimers, and the pGreen0029 ‘empty’ (negative control) plasmid vector, all verified by restriction digestion, and used to inoculate *N. benthamiana* plants. The agrobacterium cells harboring an empty vector were used as a negative control in the plant’s inoculation experiment. The experiment consisted of three treatments of ten plants and three biological replicates of each. To evaluate infectivity of the infectious clones, the two newest fully-expanded leaves (5–6 leaf stage) were agro-infiltrated [42] with (1) the DNA-A component alone, which would not be expected to infect *N. benthamiana* test/host plants (2) the ToLCNDV DNA-A + DNA-B components, which were expected to systemically infect the *N. benthamiana* test plants, and (3) the ‘empty’ plasmid vector (no insert) pGreen0029, used as the negative, mock-inoculated control. At 14 days post-inoculation (dpi), plants co-inoculated with the DNA-A + DNA-B components exhibited severe foliar curling and chlorosis, and mild stunting, confirming that the *N*. *benthamiana* plants co-infiltrated with ToLCNDV DNA-A and DNA-B infectious components were systemically infected by the virus. The negative control plants, either mock-inoculated with buffer or non-inoculated, respectively, exhibited no symptoms. Total DNA was isolated from symptomatic and asymptomatic leaves of the experimental test plants, respectively, using the CTAB method [43], followed by polymerase chain reaction (PCR) amplification to detect ToLCNDV in total DNA isolated from the test plants. The PCR reaction was carried out in Red-Taq PCR master mix (Ampliqon, Odense M, Denmark, Cat no. A180301) using the universal begomovirus primers (BegomoFor and BegomoRev), designed to amplify a ~2750 (bp) product [44] (Table 1), with 4 μL of total plant DNA as template. The no-template, negative PCR amplification control consisted of the addition of 4 μL of distilled water to the reaction mixture. The cycling parameters were as follows: initial denaturation at 98 °C for 30 s, followed by 25 cycles of denaturation at 94 °C for 60 s, annealing at 52 °C for 2 min, extension at 72 °C for 3 min, and final extension at 72 °C for 20 min. The amplicon of the expected size, respectively, was cloned into the pTX57R/T plasmid vector (Thermo Fisher Scientific, Waltham, MA, USA, Cat# K1213) and screened by colony PCR amplification, followed by confirmatory sequencing with the M13-F (forward) and -R reverse primers (Table 1).

### 2.2. Selection of Targeted Regions and Designing of gRNA Sequences for Genome Editing of ToLCNDV

The ToLCNDV DNA-A sequences (n = 346, Appendix A) available for isolates reported from Pakistan were downloaded from the NCBI database and aligned using Geneious Prime 2021.1.2021 software, available at Biomatters Ltd., Boston, MA, USA. https://www.geneious.com (accessed on 21 July 2022). Based on the alignment (Appendix A), the guide RNAs (gRNA) were designed to anneal to the coding region or open-reading frame (ORF) of the ToLCNDV (DNA-A component) coat protein (CP; AV1 ORF) spanning nucleotides 293–933, or the replication-associated protein (Rep; AC1 ORF), spanning nucleotides 1561–2324. The twenty-nucleotide-long gRNAs were selected based on several criteria, including target efficiency, specificity, and presence of a protospacer adjacent motif (PAM) sequence (N-G-G) [45] (Table 1). The next tier of gRNA selection was based on the results of off-target analysis using CRISPRdirect [46] and BLAST searches (version 1.30) from the NCBI to detect that either the chosen gRNA sequence shares sequence homology or ‘targets’ specific genome regions of the chili plant or *Nicotiana benthamiana*.

Whole sequence analysis (nanopore) of the ToLCNDV genome was carried out to confirm that the expected single nucleotide polymorphisms (SNPs) were engineered in the target regions of interest, which aimed to target regions conserved among the genome sequences of all ToLCNDV isolates available in the GenBank database. Finally, the gRNAs were evaluated for percent GC content, melting temperature, and secondary structure using Integrated DNA Technologies OligoAnalyzer software (version 3.0) (https://www.idtdna.com/pages/tools/oligoanalyzer) (accessed on 12 August 2022) (RRIR: SCR_001363), and three gRNAs for each target region were selected (Table 2) for evaluation in an in vitro cleavage assay.

### 2.3. In Vitro Cleavage Assay for Estimating the Predicted Efficiency of gRNA Cleavage

The efficiency of cleavage of four gRNAs (Table 2) designed to cleave the ToLCNDV DNA-A partial dimer ORFS, AV1 (n = 4) and AC1 (n = 2), was analyzed using the *‘*in vitro digestion Cas9 nuclease kit’ (NEB #M0386, New England Biolabs Inc., Ipswich, MA, USA), according to the manufacturer’s instructions. The gRNAs were synthesized in vitro using the ‘EnGen^®^ sgRNA synthesis Kit, S. pyogenes’ (NEB #E3322). The T7 promoter sequence (5′ TTCTAATACGACTCACTATA3′) was added at the 5′ end of the selected gRNAs, and a 14-nucleotide sequence overlapping the Cas9 scaffold (5′GTTTTAGAGCTAGA3′) was ligated to the 3′-end [27,47]. The complete oligonucleotide sequence synthesized in the in vitro gRNA reaction was 5′-TTCTAATACGACTCACTATAGNN^20^TTTTAGAGCTA-3′, where N represents the 20-nucleotide target of each gRNA (Table 3).

The sgRNA synthesis was carried out using the NEB RNA synthesis kit (#E3322), and the RNA was purified using the Monarch^®^ RNA Cleanup Kit (NEB #T2040). The quality of sgRNAs was analyzed by agarose (1%) gel electrophoresis, in 1X TAE buffer, pH 8, containing sodium hypochlorite (NaCIO) (500 µL lab bleach: 50 mL agarose). The quantity of RNA was estimated using the Qubit™ 4 Fluorometer (Thermo Fisher Scientific, Catalog #Q33238). After purification, in vitro digestion of the ToLCNDV DNA-A partial dimer component was carried out using Cas9 nuclease (NEB #M0386). The reaction contained equimolar amounts of gRNA, Cas9 protein, and the dimeric construct’s DNA as substrate (pMVL1113). The reaction mixture was incubated at 37 °C for 15 min. After incubation, the size(s) of the fragments resulting from DNA cleavage were analyzed by agarose gel electrophoresis, as described above, except without the addition of NaCIO. The plasmid DNA (pMVL1113 without the Cas9 and gRNA treatment) was included as a substrate in the negative control to evaluate the cleavage efficiency of each gRNA.

Densitometric analysis [48] was carried out by electrophoresis on a 1% agarose gel in 1X TAE buffer, pH 8, stained with ethidium bromide, and imaged on the Bio-Rad imaging system to quantify gRNA cleavage efficiency. Image Lab software (version 6.1, Tustin, CA, USA) was used to quantify band intensity from which the cleavage efficiency of each gRNA was calculated, relative to the uncut substrate DNA, by comparing the respective corresponding band intensity

### 2.4. Duplexed gRNA-Cas9 Cassette Construction

The cassettes containing the duplexed gRNA-Cas9 units were constructed according to the previously published method [49]. The cloned, duplexed gRNA-Cas9 units consisting of double-stranded gRNA sequences with flanking linkers were prepared by annealing the complementary strands. The sequences with linkers were cloned into the Golden Gate entry plasmid vectors, pYPQ131B and pYPQ132A, between the U3 or U6 promoters and gRNA scaffold sequence, to create each individual cloned gRNA (Table 4). The gRNAs 1 and 2 were designed to target the ToNDLCV AC1 ORF, while gRNAs 3 and 4 were designed to target the AV1 ORF. Whole plasmid sequencing (nanopore) was carried out to verify that each of the respective plasmid vectors contained the expected 20-nucleotide gRNA sequence.

To construct the duplexed gRNA clones, the respective pairs of gRNAs were cloned for expression under the U3 and U6 promoters in the plasmid vector, pYPQ131B and pYPQ132A, respectively, available in the Golden Gate cloning system [49]. A recombination reaction was carried out to produce a duplexed gRNA-Cas9 cassette, using the pYPQ142 and pYPQ150 plant codon-optimized Cas9 entry system, driven by the cauliflower mosaic virus 35 S promoter, and the pMDC32 plant expression (Gateway) destination vector [50]. Restriction digestion was carried out after each step to confirm the integration and accuracy of the gRNA cassettes, based on the diagnostic *Xba*I and *Eco*RV sites, to validate the Golden Gate reaction, and *Hin*dIII and *Sac*I to verify the Gateway recombination step. The resultant duplexed gRNA–Cas9 cassettes were designated pMVL1030, pMVL1031, pMVL1032, and pMVL1033, hereafter, CRISPR cassettes 1, 2, 3, and 4 (Table 5).

### 2.5. Transient Co-Inoculation of Plants with Constructs and Tomato Leaf Curl New Delhi Virus

Four duplexed gRNA-Cas9 constructs 1–4 (Table 5) and a previously constructed infectious cloned ToLCNDV DNA-A and DNA-B component genome (Brown lab, Tucson, AZ, USA; pMVL1113 & pMVL1116) were transformed separately into the disarmed strain GV3103 of Agrobacterium tumefaciens by electroporation [51]. The electroporated cells were screened on solid LB medium containing antibiotics and incubated at 28 °C for 36–48 h. A single colony resulting from each transformation reaction was selected and used to inoculate the primary culture medium, which was grown overnight as seed for the secondary culture, the latter incubated for 3–4 h to increase the inoculum to the optimal concentration of 0.6 optical density (O.D.) units. The cells were collected by centrifugation (at 4000 RPM, in an Eppendorf centrifuge machine) and resuspended in infiltration buffer, 10 mM MgCl_2_, 10 mM MES, and 150 μM acetosyringone, followed by incubation at room temperature for 3–4 h without light [42].

Seeds of N. benthamiana were sown in premixed potting medium and grown in a growth chamber with a 14:8 h light/dark cycle under LED lamps. At the 4–5 leaf stage, the transient transformation and virus co-transfection assay were carried out. The upper fully expanded leaves of each test plant were infiltrated or ‘co-inoculated’ by nicking each leaf with the tip of the syringe to create a small wound. The solution containing the respective duplexed gRNA–Cas9 construct and the ToLCNDV partial dimer was infiltrated into the leaves through the wound. Five plants were agro-inoculated with each construct (transient transformation) and the ToLCNDV infectious clone (transfection), or other treatments, in each of the three replicated experiments. Leaf samples (0.5–1.0 mg) were collected from both the infiltrated leaves and the newly developed, fully expanded leaves using a sterile leaf punch 10 days post-agroinfiltration. The negative control plants were those infiltrated with the empty pGreen0029 plasmid.

### 2.6. Detection of ToLCNDV in Agro-Infiltrated Plants by Polymerase Chain Reaction and Rolling Circle Amplification

The inoculated and non-inoculated, subsequently developing leaves of N. benthamiana plants, agro-infiltrated with the gRNA–Cas9 constructs and ToLCNDV infectious clones, were analyzed for viral accumulation. Five biological replicates and three technical replicates were analyzed for each construct. Total genomic DNA was purified from the subsequently developing leaves, 10 days post-inoculation (dpi), using the CTAB method [15]. Detection of ToLCNDV in the virus-inoculated and mock-inoculated plasmid with no insert test plants with CRISPR-Cas constructs was carried out by PCR amplification with virus-specific coat protein primers AVcore and ACcore [52] (Table 1) and previously published cycling conditions (Table 1). Circular viral molecules were amplified by rolling circle amplification (RCA) by Phi 29 DNA polymerase (Fermentas) following the standard protocol [53]. Aliquots consisting of 13 μL of Phi mixture (50 μL of 500μM hexanucleotides, 200 μL of 10 mM dNTPs, 100 μL of 10X reaction buffer, and 600 μL of ddH_2_O) were mixed with 0.5 μL of total DNA, boiled at 85 °C for 5 min, and immediately placed on ice for 5 min. The sample was incubated for 16–20 h at 30 °C after the addition of Phi 29 DNA polymerase. Following overnight incubation, fifteen μL of ddH_2_O was added to the reaction, and the Phi 29 polymerase was deactivated at 85 °C. The restricted RCA products (1 μL) were analyzed by agarose (1% *w*/*v*) gel electrophoresis in TAE buffer, pH 8.0, as described above, and the remaining RCA product was held at −20 °C.

### 2.7. Quantitative PCR Amplification and Virus Accumulation

Quantitative real-time PCR amplification was carried out to quantify the virus load in plants inoculated with the respective CRISPR construct and ToLCNDV, 10 dpi, and negative control test plants inoculated with the infectious virus clone alone and pGreen0029 empty vector. In qPCR, the actin gene was used as an internal control. The qPCR amplification reactions were carried out in the 1× KAPA SYBR^®^ FAST qPCR Master Mix (Roche, F. Hoffmann-La Roche Ltd., Basel, Switzerland). The virus load, or Cq, was determined using a standard method [54] to calculate the relative virus load in CRISPR cassette treated (10 plants treated with each cassette with three replicates) and negative control (10 plants), with virus load set at 1, for the calibrator sample consisting of *N. benthamiana* plants inoculated with ToLCNDV alone.

## 3. Results

### 3.1. Infectious Clone of ToLCNDV Isolated from Chili Plants Produced Leaf Curl Disease Symptoms in Tobacco Plants

The pathogenicity of ToLCNDV, previously cloned from chili plants in Pakistan [40], was evaluated for the co-infiltrated *N. benthamiana* plants. Co-infiltration experiments consisted of duplex gRNAs (constructs 1–4, Table 5), the bipartite, infectious DNA and DNA-B components, the infectious DNA-A component alone, or mock-inoculated *N. benthamiana* plants.

Plants were agroinfiltrated with either the ToLCNDV DNA-A and DNA-B components (wild-type combination), the DNA-A component alone, or an empty pGreen0029 vector (mock control). At 21 days post-infiltration (dpi), 8 of 10 plants inoculated with DNA-A alone showed milder symptoms of vein thickening and moderate leaf curling (Figure 1A), while all 10 plants co-inoculated with DNA-A and DNA-B developed severe symptoms, including vein thickening, pronounced downward leaf curling, and stunted plant growth (Figure 1B). No symptoms were observed in the mock-inoculated plants (Figure 1C). Total DNA was extracted from inoculated leaves using the CTAB method, and PCR amplification using universal begomovirus primers (Table 1) confirmed the presence of the DNA-A component in symptomatic plants (Figure 1D). These findings confirm that the ToLCNDV infectious clones are capable of inducing characteristic disease symptoms in *N. benthamiana*, thereby fulfilling Koch’s postulates.

### 3.2. In Vitro gRNA Cleavage of ToLCNDV Genome

An *in planta* in vitro assay was carried out to evaluate the cleavage efficiency of the gRNAs, in advance of constructing CRISPR-Cas cassettes for future transformation. Based on the highest cleavage efficiency among the thirteen gRNAs that contained the PAM motif, N-G-G, were identified within the AV1 gene between nt coordinates 293–933 (640 nt), and the AC1 gene between nt coordinates 1560–2063 (503 nt) (Table 1). Following off-target analysis against with chili pepper genome using CRISPRdirect [46] and BLASTn searches to identify single nucleotide polymorphisms (SNPs), three gRNA sequences from the latter two regions of the ToLCNDV genome were identified as having no potential off-target effects and were used in the in vitro cleavage efficiency assay (Table 1).

The in vitro cleavage assay was used to confirm the effectiveness of the selected three gRNAs from each region to cleave the ToLCNDV genome. The gRNAs from regions spanning AV1 and AC1 genes during the in vitro digestion assay showed effective cleavage of the substrate DNA (DNA-A) (Figure 2). The main objective was to apply a single and duplexed gRNA to split the viral DNA. Therefore, we employed single and duplex (using two combined gRNAs targeting both AV1 and AC1) gRNAs simultaneously to check the digested products under the in vitro conditions. The results for duplex gRNA digestion suggested that corresponding viral genomic regions can be cleaved into fragments of 6.3 kb, 2.0 kb, and 700 bp, indicating its better efficiency to cleave the viral DNA. Densitometric profiling was also performed using Image Lab software to analyze agarose gel band intensities and determine the cleavage efficiency of each gRNA compared to uncut DNA (substrate). The results showed that the cleavage efficiency of different gRNAs ranged from 45.1% to 68.6%. The Rep protein gRNA exhibited an average cleavage efficiency of 58.19%, the coat protein sequence-directed gRNA showed an average of 53.59%, and the duplexed gRNA demonstrated the highest average efficiency at 59%.

These results have demonstrated that all of the gRNA sequences employed to construct the CRISPR cassette complexes efficiently cleaved the ToNDLCV genome, singly or as duplexed gRNAs.

### 3.3. Duplexed gRNA-Cas9 Complex Vectors

The gRNAs were selected for constructing a duplexed gRNA-Cas9 cassette through Golden Gate and Gateway cloning. During the initial stage of constructing the CRISPR-Cas9 cassette, the double-stranded gRNA sequences were cloned into a Golden Gate entry vector system consisting of pYPQ131B, pYPQ132A (Table 4).

The fidelity of the cloned 20 gRNA sequences (AV1-01, AV1-02, AC1-01, AC1-03) into the Golden Gate entry vector was confirmed by DNA (Sanger) sequencing. Next, four duplex gRNA expression cassettes were developed using the Golden Gateway entry vector strategy (Figure 3A). In the third step, duplexed gRNA-Cas9 cassettes were cloned into a plant expression vector (pMDC32) (Figure 3B). Four duplexed (n = 4) CRISPR cassettes (construct 1, construct 2, construct 3, and construct 4).

Although all gRNAs showed comparable in vitro cleavage efficiency, we selected two gRNAs (AC01, AC03 and AV01, AV02) from each region to construct CRISPR cassettes. Computational predictions and initial screenings suggested that these four gRNAs had the lowest off-target potential and the highest specificity. This strategic selection was also influenced by practical constraints, such as resource availability and the feasibility of managing multiple constructs. By focusing on these four gRNAs, we ensured a robust and efficient experimental design capable of achieving our research objectives. We achieved comprehensive and non-redundant coverage of the target regions while minimizing the risk of off-target effects.

### 3.4. Efficiency of Transiently Expressed Duplex gRNA-Cas9 Complex

After constructing four duplexed gRNA-Cas9 cassettes, each containing two gRNAs targeting the AV1 and AC1 regions. The efficiency of the duplex gRNA-Cas9 complex was determined transiently in *Nicotiana benthamiana* plants using agrobacterium-mediated inoculation. Plants infiltrated with CRISPR cassettes construct 1, construct 2, construct 3, and construct 4 with ToLCNDV DNA-A and ToLCNDV DNA-B exhibited no visible symptoms of virus infection, and amplification was further confirmed by PCR and qPCR. Plants were examined for the typical symptoms after 10 days post inoculation (Table 6). The initial screenings indicated that the first two cassettes (construct 1, construct 2) exhibited higher efficiency in reducing virus accumulation and consistent expression. It shows the effectiveness of the duplex gRNA-Cas9 construct in disease attenuation. Plants inoculated with ToLCNDV DNA-A and ToLCNDV DNA-B as positive control shows severe leaf curling and stunted plant growth (Figure 4). These symptoms refer to the disruption of the plant’s natural biological processes attributed to the viral infection. The substantial leaf curling indicates the virus’s capacity to hinder the plant’s development and growth, resulting in stunted growth and lower yield, which are characteristic of geminivirus infection.

### 3.5. Duplex gRNA-Cas9 Complexes Restrict Disease Development

The viral DNA accumulation in inoculated and systemic leaves was estimated through PCR analysis. Analyzing the amplification of ToLCNDV in the gRNA-Cas9 inoculated plants by PCR enabled the evaluation of the effectiveness of gRNA-Cas9 constructs in planta. The plants treated individually with four developed CRISPR cassettes in combination with the ToLCNDV infectious clone resulted in complete symptom attenuation, with no detectable indications of leaf curling, and did not affect the plant development compared to the positive control (inoculated with ToLCNDV DNA-A and DNA-B infectious clones) plants (Figure 4). Unlike the negative control plants, none of the plants inoculated with duplexed gRNA-Cas9 exhibited leaf curl symptoms.

DNA was isolated from both inoculated and systemic leaves and subjected to PCR amplification to confirm the presence of ToLCNDV. PCR using specific coat protein primers yielded an approximately 520 bp fragment, confirming efficient viral replication in plants inoculated solely with the ToLCNDV infectious clone (Figure 5A). Similarly, plants co-inoculated with the ToLCNDV DNA-A and DNA-B infectious clones also exhibited DNA amplification, indicating the presence of the virus. Rolling Circle Amplification (RCA) further confirmed viral load in both systemic and inoculated leaves. Plants co-inoculated with ToLCNDV DNA-A and DNA-B along with CRISPR constructs 1 and 2 did not show any discernible virus symptoms, and their accumulation remained undetectable through PCR. However, a very small amount of viral DNA amplification was observed in the RCA reaction. In contrast, the constructs 3 and 4 showed visible RCA amplification on the gel (Figure 5B). Quantitative PCR (qPCR) was also performed to assess viral accumulation in infected plants. At 10 days post-inoculation (dpi), *N. benthamiana* plants infiltrated with infectious clones and CRISPR cassettes exhibited significantly reduced viral accumulation compared to control plants, which were normalized to a value of 1.00 (Figure 6).

To evaluate the viral load in plants inoculated with ToLCNDV, quantitative PCR (qPCR) analysis was performed. Plants co-inoculated with CRISPR cassettes 1 and 2, along with ToLCNDV, exhibited significantly reduced viral titers (mean ± SD: 0.09 ± 0.03 and 0.11 ± 0.02, respectively) compared to the control plants inoculated with ToLCNDV alone (1.00 ± 0.05). Plants treated with CRISPR cassettes 3 and 4 showed a moderate decrease in viral titers (0.37 ± 0.04 and 0.35 ± 0.05, respectively). Each treatment group included three biological and three technical replicates. Statistical analysis using one-way ANOVA confirmed a significant reduction in viral load for constructs 1 and 2 (*p* < 0.01). These results are consistent with the observed suppression of disease symptoms and demonstrate the effectiveness of CRISPR-Cas9 constructs in targeting ToLCNDV. Notably, CRISPR cassettes 1 and 2 led to approximately a 10-fold reduction in viral accumulation compared to the control, while constructs 3 and 4 achieved around a 3-fold decrease. These findings highlight the superior performance of constructs 1 and 2 in reducing ToLCNDV replication and underscore the potential of the gRNA-Cas9 approach as a targeted antiviral strategy.

## 4. Discussion

The goal of this study was to explore the potential of duplexed CRISPR-Cas9 technology in controlling Tomato Leaf Curl New Delhi Virus (ToLCNDV), a major threat to solanaceous crops in several parts of the world [12,19,21,23,55,56], including the Indian sub-continent. Our findings provide compelling evidence that targeting the ToLCNDV genome using duplex guide RNAs (gRNAs) targeting two different coding regions (coat protein and *rep* genes) effectively reduces viral accumulation and mitigates symptom expression in *Nicotiana benthamiana.* The experiments represent a significant advancement toward utilizing genome-editing tools for engineering resistance to plant begomovirus infection. Given the substantial negative effects of ToLCNDV on economically important crops such as chili pepper, cucurbits, and tomato, this underscores the need for virus resistance in susceptible crop plants to reduce the damage caused by ToLCNDV infection. Because traditional breeding efforts have not succeeded in developing resistant cultivars, alternative methods are needed.

The CRISPR-Cas9 editing system [57], which brings editing precision and flexibility, [37,58] offers a promising path forward. By targeting two essential begomoviral genes, *rep* and *cp*, both virus replication, movement, and potentially, encapsidation were significantly suppressed, in turn mitigating disease symptoms in the virus-infected tobacco plants, compared to the negative control plants.

The results have demonstrated that dual targeting of two begomoviral genes using a duplexed gRNA strategy effectively attenuates Tomato leaf curl New Delhi virus (ToLCNDV) infection in *Nicotiana benthamiana*. Constructs 1 and 2, which simultaneously targeted two essential viral genes, resulted in reducing viral accumulation and suppressing symptom development. In contrast, constructs 3 and 4, which targeted a single viral gene each—the coat protein and replication-associated protein (Rep) genes, respectively—were less effective. The observed reduction in viral accumulation with constructs 1 and 2 (dual-target) aligns with previous reports suggesting duplexed gRNAs may reduce escape mutants. Plants treated with constructs 1 and 2 showed approximately 10-fold lower viral accumulation, as confirmed by qPCR, and did not display characteristic ToLCNDV symptoms. This suggests a substantial attenuation of viral virulence. Conversely, constructs 3 and 4 resulted in relatively higher viral loads and a less pronounced reduction in symptom severity, with statistical analysis showing non-significant differences (*p* > 0.05) compared to the highly significant reductions seen with constructs 1 and 2 (*p* < 0.01). These findings are consistent with previous reports demonstrating the enhanced efficacy of duplexed CRISPR-Cas9 approaches in achieving robust viral interference [36]. Furthermore, rolling circle amplification (RCA) analysis supported these observations: while ToLCNDV-infected plants showed strong RCA signals, those co-inoculated with the virus and CRISPR constructs showed no detectable viral DNA, indicating effective suppression of viral replication.

Several previous studies have demonstrated the effectiveness of CRISPR-Cas9 in combating plant viruses, including begomoviruses such as *Tomato yellow leaf curl virus* (TYLCV) and *Beet curly top virus* (BCTV). However, these studies often employed single-guide RNA (gRNA) constructs, which, while somewhat effective, are prone to the emergence of escape mutants that evade CRISPR targeting [37,59]. The approach used in this study, involving duplexed gRNA constructs, was designed to overcome these limitations. Constructs 3 and 4, which targeted a single viral region, resulted in relatively higher virus accumulation compared to constructs 1 and 2, which targeted two distinct regions in the viral genome. This suggests that even duplexed gRNAs may be ineffective if they target only a single region. By simultaneously targeting two essential viral genes, we minimized the likelihood of escape mutants, thereby providing a more robust and durable resistance strategy.

Previous studies have demonstrated the genome editing efficiency of the CRISPR-Cas9 system against Mesta yellow vein mosaic virus and Cotton leaf curl Khokhran virus and evaluated its performance when targeting different coding and non-coding regions of viral genomes. These reports showed that targeting coding regions often led to escape mutants, which interfered with CRISPR activity and reduced its effectiveness. Mehta et al. (2019) reported that using CRISPR-Cas technology in cassava to target the African cassava mosaic virus resulted in new viral variants, ultimately failing to confer effective resistance [37]. In this context, Rybicki and colleagues critically analyzed those findings and concluded that targeting one virus target with a single gRNA was likely responsible for the formation of escape mutants, hence, simultaneous targeting of multiple viral genome regions using two or more gRNAs, coupled with stable Cas9 expression, should significantly reduce the likelihood of escape mutants. Supporting this, earlier studies have indicated that escape mutants can be avoided by targeting multiple distinct sites in the viral genome simultaneously [38]. Recent findings further strengthen the concept that single-site targeting may be unreliable for durable virus restriction, as it often promotes mutant virus survival [37,58]. One of the most promising aspects of the CRISPR-Cas9 system is its capacity to incorporate multiple gRNAs into a single construct, allowing the simultaneous targeting of different viral genes [60,61]. For instance, Baltes et al. demonstrated that the use of two distinct gRNA cassettes, targeting separate viral genes and inoculated together, significantly reduced viral titers compared to single gRNA applications [35]. In a recent study, a duplexed gRNA-based CRISPR-Cas9 approach was evaluated for targeting chili leaf curl virus (ChiLCV) at multiple genomic loci, simultaneously. The results showed that virus infection was abated, and no mutants were detected that could have potentially escaped. This study was the first to demonstrate that duplexed CRISPR-based targeting can offer robust resistance against begomoviruses, highlighting its potential in plant viral defense [53]. These findings align with these previous reports, supporting the conclusion that a duplexed gRNA toolbox [49] is more robust and effective than using a single CRISPR-Cas9 cassette.

The transient expression and co-inoculation approach evaluated in this study allowed for the preliminary in planta evaluation of multiple CRISPR-Cas9 constructs. While transient expression is a valuable tool for conducting proof-of-concept studies [39], stably transformed plants that express gRNA-Cas9 continuously will be required to evaluate the effectiveness and durability of virus resistance in the long term [37,59]. It has been demonstrated that CRISPR-Cas9-mediated mutations can be stably inherited in crops such as rice and tobacco, suggesting that the strategy under development here could be applied to other economically important crop plant species that are hosts of ToLCNDV [35,59]. Also, the off-target analysis employed in gRNA design suggested that the gRNAs selected in this study would have minimal risk of disrupting regions of the genome that were not targeted in the plant host. This step in gRNA design is critical for ensuring the fidelity and target-specificity of CRISPR-Cas activity, particularly when considering regulatory approval for genetically modified crops. One caveat is that this study was carried out using *N. benthamiana*, a plant model system commonly used for plant–virus interaction studies because it is highly susceptible to infection by most plant viruses. While the results obtained for *N. benthamiana* are valid, the feasibility of CRISPR-Cas technology for virus disease management will require the development of stably modified crop plant species to evaluate the efficacy and specificity of the approach for large-scale application to virus control in agricultural systems.

## 5. Conclusions

In summary, the results reported here demonstrate that CRISPR-Cas9 technology shows promise for mitigating ToLCNDV infection, particularly when targeting multiple viral regions. We achieved significant reductions in viral accumulation and disease symptoms by targeting two critical regions of the viral genome with duplexed gRNAs. These findings represent an important step toward developing transgenic crops with durable resistance to ToLCNDV, contributing to food security and sustainable agricultural practices. Future work will focus on translating these findings to crop species and further refining the technology for large-scale deployment.

## Figures and Tables

**Figure 1 pathogens-14-00679-f001:**
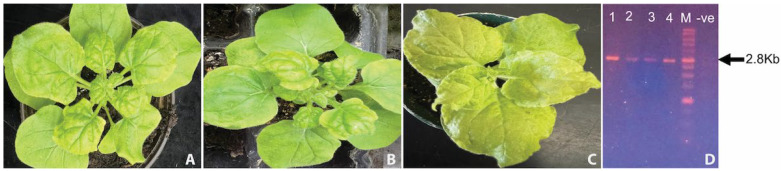
Agrobacterium-inoculation of *Nicotiana benthamiana* plants: (**A**) plants inoculated with the tomato leaf curl New Delhi virus (ToLCNDV) DNA-A component cloned into the pGreen0029 plasmid vector; (**B**) leaf curling and vein-banding symptoms upon post-inoculation of *N. benthamiana* plants with ToLCNDV DNA A and DNA B components into the pGreen0029 plasmid vector; (**C**) plants mock-inoculated with the empty plasmid vector, pGreen0029; (**D**) the PCR products amplified from test plants. The lane marked as M contains the 1 kbp ladder (Thermo Scientific). The arrow indicates a band size of ~2.8 Kb. Lane 1–4 contain the expected ~2750 base pair fragment of the ToLCNDV (DNA-A genome), amplified from the plants agro-inoculated with the ToLCNDV DNA-A and B component partial dimer. The lane marked negative control, –ve, contains the reaction mix without the template.

**Figure 2 pathogens-14-00679-f002:**
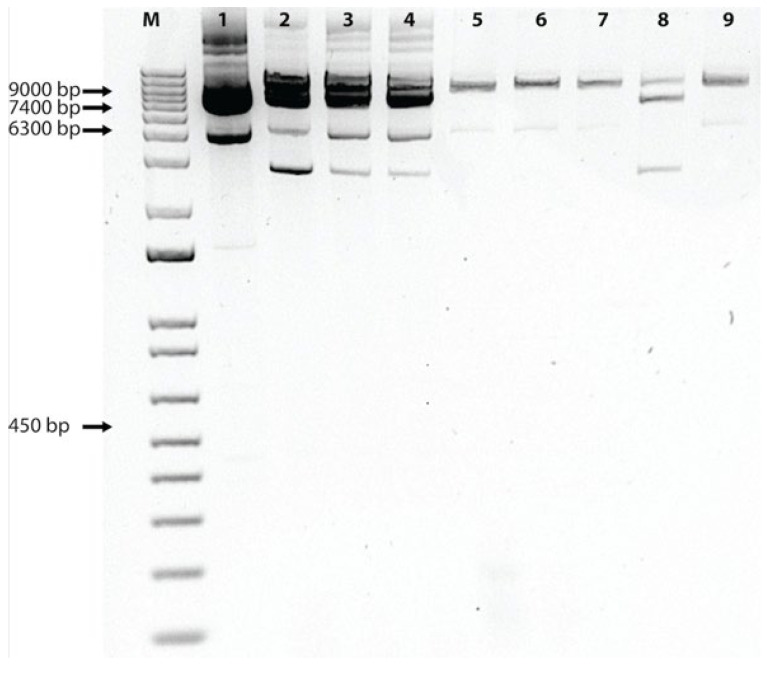
In vitro cleavage of DNA with gRNA-Cas9 complex. M represents 1 Kb plus ladder, Lane-1 shows uncut viral genome (pMVL 1113) without the gRNA and Cas9 treatment. Lanes 2–4 show the rep region targeted by the gRNAs, with an average cleavage efficiency of 58.19%, (AC1-01, AC1-02, and AC1-03). Lanes 5–7 indicate CP region gRNA (AV1-01, AV1-02, AV1-07), representing average cleavage efficiency of 53.59%. Lanes 8–9 show activities of duplex gRNA with an average cleavage efficiency of 59%.

**Figure 3 pathogens-14-00679-f003:**
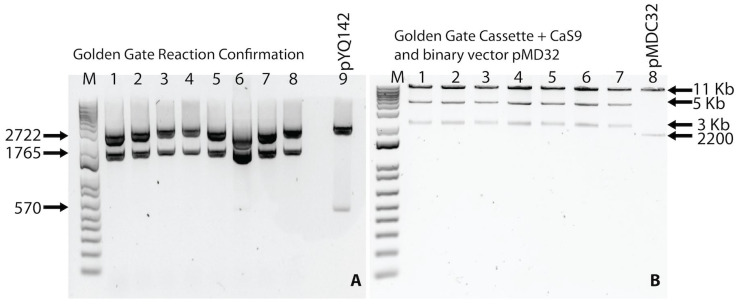
Golden gate and gateway reaction confirmation. (**A**) Lane 1–8 shows restriction digestion for the confirmation of the Golden Gate reaction using *Eco*RV and *Xba*I enzymes. Lane 9 indicates restriction of the empty pYPQ142 plasmid (Golden gate recipient vector) with *Eco*RV and *Xba*I. (**B**) Lane 1–7 shows restriction digestion for confirmation of the gateway reaction (containing Golden Gate reaction cassette, Cas9, and binary vector) with *Sac*I and *Kpn*I. Lane 8 shows the restriction digestion product of the binary vector pMDC32 containing no cloned insert. M represents 1 Kbp Plus DNA marker.

**Figure 4 pathogens-14-00679-f004:**
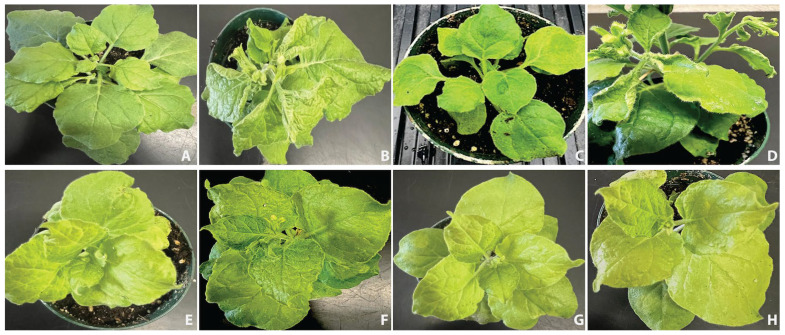
Inoculation of *N. benthamiana* with CRISPR cassettes: (**A**) healthy *Nicotiana benthamiana* plant (Mock); (**B**) ToLCNDV DNA-A dimer; (**C**) ToLCNDV DNA-B dimer; (**D**) ToLCNDV DNA-A and DNA-B dimer (positive control); (**E**) CRISPR construct 1 with ToLCNDV dimer; (**F**) CRISPR construct 2 in association with ToLCNDV dimer; (**G**) CRISPR construct 3 in association with ToLCNDV dimer; (**H**) CRISPR construct 4 in association with ToLCNDV partial dimer.

**Figure 5 pathogens-14-00679-f005:**
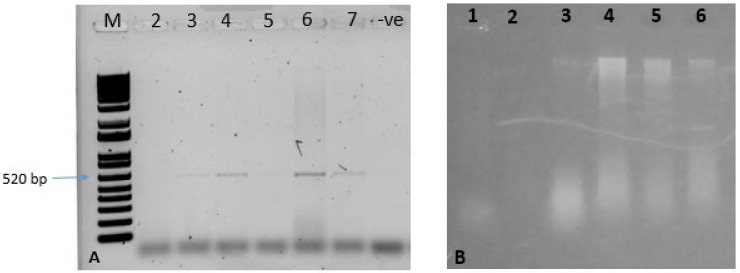
Confirmation of ToLCNDV in plants by PCR and RCA: (**A**) M represents 1 Kb plus ladder, Lane 2–5 shows amplification by core coat protein primers of the plants treated with CRISPR constructs and ToLCNDV dimer, Lane 6–7 shows the amplification of the plants treated with only ToLCND virus DNA-A and DNA-B; (**B**) Lane 1–4 shows the RCA amplification of plants treated with CRISPR cassettes and infectious clones, Lane 5–6 shows the plants only treated with infectious clones of ToLCNDV.

**Figure 6 pathogens-14-00679-f006:**
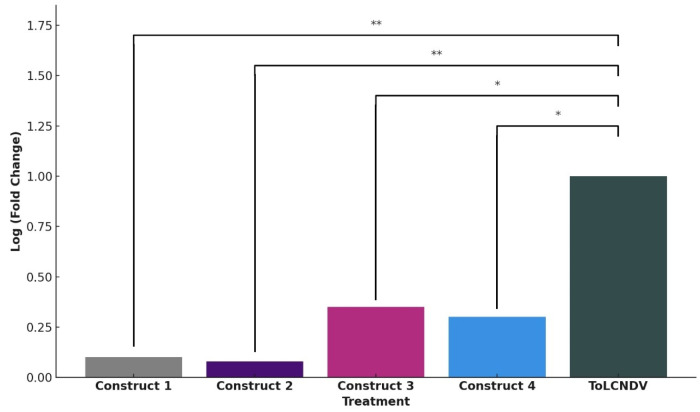
qPCR to detect viral accumulation in plants. Viral accumulation in plants treated with dual-target (constructs 1/2) or single-target (constructs 3/4) gRNAs. The relative accumulation of ToLCNDV in *N. benthamiana* plants at 10 dpi is shown. Plants co-infiltrated with the infectious clone (ToLCNDV) and CRISPR-Cas constructs are represented by the respective colored bars for constructs 1–4. In contrast to the positive control, plants inoculated with the infectious ToLCNCV cloned DNA-A and DNA-B components showed minimal viral accumulation at * *p* < 0.01 or ** *p* < 0.05. indicative of a significant difference by ANOVA. Constructs 1 and 2, targeting two viral regions, showed reduced viral accumulation compared to constructs 3 and 4 (single-target).

**Table 1 pathogens-14-00679-t001:** Primers used for polymerase chain reaction (PCR) amplification of tomato leaf curl New Delhi virus and for colony PCR amplification, as described.

Primer Name	Primer Sequence
BegomoF	5′ GCCYATRTAYAGRAAGCCMAG 3′
BegomoR	5′ GGRTTDGARGCATGHGTACANGCC 3′
pTC14-F2	5′ CAAGCCTGATTGGGAGAAAA 3′
M13-F	5′ CCCAGTCACGACGTTGTAAAACG 3′
M13-R	5′ GGAAACAGCTATGACCATG 3′
AVcore	5′ GCCHATRTAYAGRAAGCCMAGRAT 3′
ACcore	5′ GGRTTDGARGCATGHGTACANGCC 3′

**Table 2 pathogens-14-00679-t002:** Details of the gRNA sequences selected from conserved regions of representative ToLCNDV genome sequences, and results of single nucleotide polymorphisms (SNPs) and off-target analyses.

Genomic Region	gRNA Name	Sequence	PAM	Position in Genome	No. of SNP’s	CRISPR-Direct		
						20 mer PAM	12 mer PAM	8 mer PAM
AV1 ORF (Coat Protein region)	AV1-01 *	GTTGAAATGATGATATCTGC *	TGG	293- 315	1	0	0	517
	AV1-02 *	GTTCATCGGCCTGTTGGTCC *	AGG	459–481	1	0	0	176
	AV1-03	TAGAAGTCCCGACGTGCCAA	GGG	460–482	4	0	0	152
	AV1-04	TGTGTTAGTGATGTTACCCG	AGG	554–576	4	0	1	55
	AV1-05	GTGAAATCCGTCTATGTGCT	GGG	614–636	10	0	2	164
	AV1-06	GAAGTGGCATGCGACTGTGA	CGG	823–845	7	0	0	1020
	AV1-07 *	GTTATCAAGTCTTACGGAAG *	TGG	807–829	2	0	0	416
	AV1-08	GTTTATAATCAACAAGAGGC	CGG	911–933	5	0	0	231
AC1 ORF (Rep Protein region)	AC1-01 *	CTCGAAGAACCACTCTATTC *	CGG	1561–1583	1	0	3	436
	AC1-02 *	GATTTAGCTCCCTGAATGTT *	CGG	2302–2324	1	0	1	2864
	AC1-03 *	GTCATCAATGACGTTGTACC *	AGG	1803–1825	0	0	1	536
	AC1-04	GGGCCTAAAAGGCCGCGCAG	CGG	1947–1969	7	0	0	72
	AC1-05	GGAGAAACATAAACCTCGGA	AGG	2041–2063	12	0	0	38

The asterisk * indicates the sequences selected for the in vitro assay on the basis of minimum no of SNP’s and off target effects from CRISPR direct. The number of single nucleotide polymorphisms (SNPs) in each gRNA sequence was determined for the genome sequence of all available Pakistan isolates of ToLCNDV. The gRNA sequences were analyzed using CRISPRdirect and documented for all 8, 12, and 20 mers.

**Table 3 pathogens-14-00679-t003:** Nucleotide sequences used for the synthesis of gRNAs in the in vitro cleavage assay. The blue and red letters indicate the T7 promoter region and gRNA sequence, respectively, while the black letters indicate the 14-nucleotide overlapping region.

gRNA	Sequence from 5′ to 3′
AV1-01	TTCTAATACGACTCACTATAGTTGAAATGATGATATCTGCGTTTTAGAGCTAGA
AV1-02	TTCTAATACGACTCACTATAGTTCATCGGCCTGTTGGTCCGTTTTAGAGCTAGA
AV1-07	TTCTAATACGACTCACTATAGTTATCAAGTCTTACGGAAGGTTTTAGAGCTAGA
AC1-01	TTCTAATACGACTCACTATAGCTCGAAGAACCACTCTATTCGTTTTAGAGCTAGA
AC1-02	TTCTAATACGACTCACTATAGATTTAGCTCCCTGAATGTTGTTTTAGAGCTAGA
AC1-03	TTCTAATACGACTCACTATAGTCATCAATGACGTTGTACCGTTTTAGAGCTAGA

**Table 4 pathogens-14-00679-t004:** Sequences of double-stranded gRNA oligonucleotides and linkers, and the corresponding Golden Gate entry vectors and cloning sites used to produce the gRNA constructs. The red letters indicate the position and sequence of the linkers.

Selected gRNA	Two Strands of Oligo Sequence with Linkers (5′-3′)	Entry Vector	Name of Clone
AC1-03	Top: GATTGTCATCAATGACGTTGTACC	pYPQ131	gRNA 1
	Bottom: AAACGGTACAACGTCATTGATGAC		
AC1-02	Top: GATTGATTTAGCTCCCTGAATGTT	pYPQ132	gRNA 2
	Bottom: AAACAACATTCAGGGAGCTAAATC		
AV1-02	Top: GATTGTTCATCGGCCTGTTGGTCC	pYPQ132	gRNA 3
	Bottom: AAACGGACCAACAGGCCGATGAAC		
AV1-01	Top: GATTGTTGAAATGATGATATCTGC	pYPQ131	gRNA 4
	Bottom: AAACGCAGATATCATCATTTCAAC		

**Table 5 pathogens-14-00679-t005:** CRISPR-Cas9 constructs and the respective Tomato leaf curl New Delhi virus genome target.

Construct Name	gRNA Sequence	Target Region
Construct 1	gRNA 1 + gRNA 3	Rep region and CP region (Dual target)
Construct 2	gRNA 4 + gRNA 2	Rep region and CP region (Dual target)
Construct 3	gRNA 4 + gRNA 3	CP region (Single target)
Construct 4	gRNA 1 + gRNA 2	Rep region (Single target)

**Table 6 pathogens-14-00679-t006:** Summary of treatment groups and plant counts.

Treatment	No of Plants Replication-1	No of Plants Replication-2	No of Plants Replication-3	Symptoms
CRISPR construct 1 coupled with ToLCNDV DNA-A & DNA-B dimer	10 plants	10 plants	10 plants	No symptoms
CRISPR construct 2 coupled with ToLCNDV DNA-A & DNA-B dimer	10 plants	10 plants	10 plants	No symptoms
CRISPR construct 3 coupled with ToLCNDV DNA-A & DNA-B dimer	10 plants	10 plants	10 plants	No symptoms
CRISPR construct 4 coupled with ToLCNDV DNA-A & DNA-B dimer	10 plants	10 plants	10 plants	No symptoms
ToLCNDV DNA-A & DNA-B dimer (negative control)	10 plants	9 plants	9 plants	Severe leaf curling and vein thickening
ToLCNDV DNA-A dimer	5 plants	5 plants	5 plants	Severe leaf curling and vein thickening
Mock control (inoculated with non-transformed agrobacterium)	10 plants	10 plants	10 plants	No symptoms

## Data Availability

The original contributions presented in this study are included in the article/Appendix A. Further inquiries can be directed to the corresponding author.

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
