# Peer review of "Potential for Duplexed, In-Tandem gRNA-Mediated Suppression of Two Essential Genes of Tomato Leaf Curl New Delhi Virus in Crop Plants"

_pathogens, 2025, doi:10.3390/pathogens14070679_

Round 1
Reviewer 1 Report
Comments and Suggestions for Authors
The paper describes the use of CRISPR-Cas9 to control a DNA virus infection.
- The authors state that the use of "multiplexed gRNAs" (which are actually two gRNAs) is a novel, very efficient approach. Therefore, it is necessary to demonstrate that the simultaneous use of two gRNAs results in a better efficiency of virus suppression than the use of a single gRNA. Thus, additional experiments are needed to prove the main point that the authors make in this paper.
- The amount of experimental work reported is not sufficient for a high impact journal such as Pathogens. In fact, in addition to construct design and verification, the paper reports a single experiment on co-agroinfiltration of CRISPR-Cas9/gRNA constructs with ToLCNDV infectious clones described in Sections 3.4 and 3.5.
- The writing of the paper does not meet the standards accepted in the scientific literature. Each subsection of Results should state (1) what the purpose of the experiments was, (2) how the experiments were performed, (3) what the results were, and (4) what the results mean. This is not the case in the paper by Naveed et al. In many cases, it is not clear what experiments were done and what the results were. In addition, experimental work should not be described in the Materials and Methods section. Therefore, the paper should be completely rewritten before resubmission.
Author Response
Reviewer #1:
The paper describes the use of CRISPR-Cas9 to control a DNA virus infection.
Q1: The authors state that the use of "multiplexed gRNAs" (which are actually two gRNAs) is a novel, very efficient approach. Therefore, it is necessary to demonstrate that the simultaneous use of two gRNAs results in a better efficiency of virus suppression than the use of a single gRNA. Thus, additional experiments are needed to prove the main point that the authors make in this paper.
Ans: We appreciate the reviewer’s concern regarding the difference between results of one guide RNA vs 2 gRNAs. Indeed prior to conducting these experiments, the literature suggested that singe gRNA may be insufficient to confer resistance against geminiviruses. For example, experiments conducted by Ali et al., on cotton leaf curl Kokhran virus and experiments done by Mehta et al., on African cassava mosaic virus demonstrated the existence of escape mutants in the presence of single gRNA. Later, in this context, Rybicky and his colleagues hypothesized that in order to take full advantage of CRISPR/CaS based engineered resistance single gRNA may not be sufficient to confer resistance against geminiviruses. Recent studies by Baltes et al., also suggested the use of multiple guide RNAs to reduce the viral loads instead of single guide RNAs. Therefore, we believe that double guide RNA experiment is appropriate and single guide RNA is insufficient and introduces viral mutants, which can escape the resistance strategy. As a reference we provide here the literature, which we consulted while designing the experiments.
- Ali, Z., S. Ali, M. Tashkandi, S. S.-e.-A. Zaidi and M. M. Mahfouz (2016). "CRISPR/Cas9-mediated immunity to geminiviruses: differential interference and evasion." Sci. Rep. 6(1): 26912.
- Baltes, N. J., A. W. Hummel, E. Konecna, R. Cegan, A. N. Bruns, D. M. Bisaro and D. F. Voytas (2015). "Conferring resistance to geminiviruses with the CRISPR–Cas prokaryotic immune system." Nature Plants 1(10): 1-4.
- Mehta, D., A. Sturchler, R. B. Anjanappa, S. S.-e.-A. Zaidi, M. Hirsch-Hoffmann, W. Gruissem and H. Vanderschuren (2019). "Linking CRISPR-Cas9 interference in cassava to the evolution of editing-resistant geminiviruses." Genome Biol. 20: 1-10.
- Rybicki, E. P. (2019). "CRISPR–Cas9 strikes out in cassava." Nature Biotechnology 37(7): 727-728.
Q2: The amount of experimental work reported is not sufficient for a high impact journal such as Pathogens. In fact, in addition to construct design and verification, the paper reports a single experiment on co-agroinfiltration of CRISPR-Cas9/gRNA constructs with ToLCNDV infectious clones described in Sections 3.4 and 3.5
Ans: We thank the reviewer for their valuable feedback. While we understand the expectation for broader experimental scope in high-impact publications, we would like to clarify that our study is intended as a proof-of-concept to demonstrate the potential of CRISPR-Cas9-mediated resistance against ToLCNDV. Rather than expanding the dataset excessively, we focused on establishing foundational evidence with clarity and scientific rigor. In this manuscript, we first constructed infectious clones of ToLCNDV DNA-A and DNA-B. Next, we designed and validated guide RNAs (gRNAs) targeting key regions of the viral genome. These gRNAs were tested for efficiency in vitro, and based on those results, we developed four CRISPR-Cas9 constructs to evaluate their ability to attenuate viral symptoms in plants. To assess viral suppression, we measured viral titers using quantitative PCR, rolling circle amplification (RCA), and coat protein gene-specific primers. This multi-pronged approach confirms the effectiveness of our CRISPR strategy and underscores its promise as a potential biotechnological tool for controlling Tomato leaf curl New Delhi virus (ToLCNDV). We believe this focused study lays essential groundwork for future research and more extensive field-level validation.
Q3: The writing of the paper does not meet the standards accepted in the scientific literature. Each subsection of Results should state (1) what the purpose of the experiments was, (2) how the experiments were performed, (3) what the results were, and (4) what the results mean. This is not the case in the paper by Naveed et al. In many cases, it is not clear what experiments were done and what the results were. In addition, experimental work should not be described in the Materials and Methods section. Therefore, the paper should be completely rewritten before resubmission.
Ans: We appreciate the reviewer’s feedback and believe that the revised manuscript now presents our findings in a more rigorous and accessible manner. We recognize that each subsection should clearly describe (1) the purpose of the experiment, (2) how it was performed, (3) the key results, and (4) their interpretation. In the revised manuscript, we have carefully restructured each Results subsection to follow this standard scientific format, thereby improving clarity, coherence, and alignment with accepted publication practices. Additionally, we have reviewed the Materials and Methods section to ensure that procedural descriptions are appropriately placed and do not overlap with the presentation of results.
Reviewer 2 Report
Comments and Suggestions for Authors
After a thorough review of the manuscript, I have identified some points that need to be addressed before the manuscript can be considered for publication. These suggestions aim to improve the clarity, accuracy, and overall quality of the paper. Below, I have outlined the necessary revisions:
Iine 9: in this and other references for the virus. Please attend to the new viral binomial nomenclature and incorporate it in the citation on the manuscript. Can be checked and confirmed on the ICTV website.
Line 30: have
Line 39: What do you mean by developmental stage? vegetative, fruit, flowering? Perhaps add “early developing stages”
Line 53- 59: Is there any statistical data on the estimation of loss that can be added here?
Line 99-122: For clarity, consider adding a diagram or plasmid map if possible to visually support this multistep clonings.
Line 115: were used to inoculate Nicotiana benthamiana
Line 115: remove Where
Line 118: Fourteen days post-inoculation
Line 121: begomovirus primers
Line 122: consider adding information of the controls used on the PCR reaction
Line 125-126: Specify the number of sequences aligned
Line 127-130: For clarify purposes, mention why these regions were selected
Line 218-226: Clarify if technical and/or biological replicates were included
Line 243-244: Clarify whether PCR results were confirmed by sequencing or whether only amplification was used as a diagnostic
Line 248: Fix (C)
Line 295: Although all gRNAs showed comparable in vitro cleavage efficiency
Line 305: After constructing four multiplexed gRNA-Cas9 cassettes, each
Line 309: no visible symptoms
Line 310: clarify if PCR or qPCR was used to confirm the absence of viral DNA.
Line 313: how the "consistent expression" was measured?
Line 304-319: It would be helpful to report viral load data from qPCR to support the symptom observations, even if this number does not show infection, it would make the findings clearer and strongly supported.
Line 320, figure 4: consider redimension of D, it looks like a little distorted. On F, top leaves looks a little curling, perhaps change the picture to one that will better show what is stated on table 6.
Line 330: it was possible to determine the effectivity” change to “enabled the evaluation of effectiveness”
Line 343-344: Clarify whether PCR and RCA were conducted on both inoculated and systemic tissues in each treatment group.
Line 348: Missing clear numeric results of the qPCR assays. Add the exact fold reduction in viral load from qPCR to strengthen your conclusion of attenuation. Although present on figure 6, stating it on the main text will help better understand.
Line 366: The main
Line 367: again nomenclature style problem, please revise it on the entire text.
Line 369: change results to findings
Line 370-371: down-regulates viral accumulation and symptoms 370 in Nicotiana benthamiana plants change to reduces viral accumulation and mitigates symptom expression in Nicotiana benthamiana
Line 371: remove conducted here
Line 372: for engineering resistance to plant viruses
Line 373: change “Major impact of ToLCNDV on economically important crops like “ to Given the substantial impact of ToLCNDV on economically important crops such as.
Line 375-376: down-tone “making it essential to explore other approaches” to highlighting the necessity of exploring alternative methods
Line 376-377: Same problem “CRISPR-Cas9, known for its precision and adaptability [4, 19, 33], offers a promising path forward”
Line 394-395: viral nomemclature – please fix it
Line 396: replace "prone" with susceptible
Can be add on the introduction due to their relevance: Zhan et al 2022, 2024 https://doi.org/10.3390/plants13233313
https://doi.org/10.3390/ijms23042303
These references offer significant insights into CRISPR-based strategies for plant-virus resistance and are good reviews for this topic.
Once these revisions have been incorporated, I believe the manuscript will be much stronger for publication. I congratulate the authors on their innovative approach and the potential impact of their work on plant viral resistance.
Author Response
Reviewer-2
Comments and Suggestions for Authors
After a thorough review of the manuscript, I have identified some points that need to be addressed before the manuscript can be considered for publication. These suggestions aim to improve the clarity, accuracy, and overall quality of the paper. Below, I have outlined the necessary revisions:
Q1- Iine 9: in this and other references for the virus. Please attend to the new viral binomial nomenclature and incorporate it in the citation on the manuscript. Can be checked and confirmed on the ICTV website.
Ans: We thank the reviewer for pointing this out. In accordance with the updated ICTV guidelines on virus species nomenclature, we have revised the manuscript to reflect the correct binomial species name for Tomato leaf curl New Delhi virus. All relevant instances in the manuscript have been updated to follow the standardized format. We have also crosschecked the nomenclature using the ICTV Master Species List to ensure accuracy.
Q2- Line 30: have
Ans: We replaced ‘including’ with ‘have’ as suggested.
Q3- Line 39: What do you mean by developmental stage? vegetative, fruit, flowering? Perhaps add “early developing stages”
We agree that the term “developmental stage” was too vague. We have revised the text to specify “during early growth stages” which provide more clarity. This change has been incorporated.
Q4- Line 53- 59: Is there any statistical data on the estimation of loss that can be added here?
a line was added in line 55 and 56. “and at least forty plants species are hosts of ToLCNDV [18] of which some may suffer 100% yield loss [19].”
Q5- Line 99-122: For clarity, consider adding a diagram or plasmid map if possible to visually support this multistep cloning.
Answer: Thank you for the suggestion. We agree that a visual representation would enhance clarity. A plasmid map illustrating the multistep cloning strategy has now been added as supplementary figure.
Q6- Line 115: were used to inoculate Nicotiana benthamiana
Ans: Now line 125. The line has been corrected as follows “and used to inoculate N. benthamiana plants.”
Q7- Line 118: Fourteen days post-inoculation
Ans: Now line 133, The phrase ‘After 14 days’ post inoculation’ has been revised to “At 14 days post-inoculation (dpi)”
Q8- Line 121: begomovirus primers
Ans: Now line 142, The phrase ‘universal primers of begomovirus (BegomoF & BegomoR)’ has been revised to ‘with universal begomovirus primers (BegomoFor and BegomoRev) ’
Q9- Line 122: consider adding information of the controls used on the PCR reaction
Ans: Now line 144, is updated as “The negative control consisted of the addition of 4 μL of distilled water e.g. no template to the reaction mixture. “
Q10- Line 125-126: Specify the number of sequences aligned
Ans: We aligned all the available sequences of ToLCNDV DNA-A genome reported from Pakistan (n = 346)
Q11- Line 127-130: For clarify purposes, mention why these regions were selected
Ans: Line 87 to 89. “The AV1 gene is involved in encapsidation of the viral genome and is essential for cell-to-cell movement within the host, while AC1 is essential for the initiation of viral replication.”
This information is updated in the introduction section.
Q12- Line 218-226: Clarify if technical and/or biological replicates were included
Ans: We confirm that both biological and technical replicates were included in the experiments. Specifically, each assay was performed using three biological replicates (independent plants) and three technical replicates (repeated measurements from the same sample) to ensure accuracy and reproducibility. This information has now been clearly stated in the Materials and Methods section.
Q13- Line 243-244: Clarify whether PCR results were confirmed by sequencing or whether only amplification was used as a diagnostic
Ans: PCR amplification was used as a diagnostic method, along with observation of characteristic viral symptoms in the infected plants. Sequencing was not performed for PCR confirmation.
Q14- Line 248: Fix (C)
Ans: We have corrected the labeling of panel (C) in Figure [1]
Q15- Line 295: Although all gRNAs showed comparable in vitro cleavage efficiency
Ans: The phrase has been revised to ‘The gRNAs from regions spanning AV1 and AC1 genes during the in vitro digestion assay showed effective cleavage of the substrate DNA (DNA-A) “
Q16- Line 305: After constructing four multiplexed gRNA-Cas9 cassettes, each
Ans: The phrase has been revised to “We constructed four multiplexed CRISPR cassettes (construct 1, construct 2, construct 3 and construct 4) at the end of the gateway cloning.”
Q17- Line 309: no visible symptoms
Ans: The phrase ‘no symptoms of virus infection’ has been revised to ‘no visible symptoms of virus infection’.
Q18- Line 310: clarify if PCR or qPCR was used to confirm the absence of viral DNA.
Ans: 424-425, “The viral DNA accumulation in inoculated and systematic leaves was estimated through PCR analysis.”
444-445, “Quantitative PCR (qPCR) was also performed to assess viral accumulation in infected plants”
Q19- Line 313: how the "consistent expression" was measured?
Ans: The expression results in measurable effect (e.g., symptoms suppression), because we repeated the experiment three times and same result were achieved in multiple plants.
Q20- Line 304-319: It would be helpful to report viral load data from qPCR to support the symptom observations, even if this number does not show infection, it would make the findings clearer and strongly supported.
Ans: To evaluate the viral load in plants inoculated with ToLCNDV, quantitative PCR (qPCR) analysis was performed. Plants co-inoculated with CRISPR cassettes 1 and 2, along with ToLCNDV, exhibited significantly reduced viral titers (mean ±â€¯SD: 0.09 ±â€¯0.03 and 0.11 ±â€¯0.02, respectively) compared to the control plants inoculated with ToLCNDV alone (1.00 ±â€¯0.05). Plants treated with CRISPR cassettes 3 and 4 showed a moderate decrease in viral titers (0.37 ±â€¯0.04 and 0.35 ±â€¯0.05, respectively). Each treatment group included three biological and three technical replicates. Statistical analysis using one-way ANOVA confirmed a significant reduction in viral load for constructs 1 and 2 (P < 0.01).
Q21- Line 320, figure 4: consider redimension of D, it looks like a little distorted. On F, top leaves looks a little curling, perhaps change the picture to one that will better show what is stated on table 6.
Ans: Thank you for your observation. We have redimensioned panel D in Figure 4 to correct the visual distortion and improve clarity. Additionally, we have replaced panel F with a new image that more clearly shows the leaf morphology described in Table 6, particularly reducing any ambiguity regarding curling symptoms in the top leaves.
Q22- Line 330: it was possible to determine the effectivity” change to “enabled the evaluation of effectiveness”
Ans: The phrase ‘it was possible to determine the effectivity of multiplexed gRNA-Cas9 constructs in planta.’ Has been revised with ‘ enabled the evaluation of effectiveness of multiplexed gRNA-Cas9 constructs in planta”
Q23- Line 343-344: Clarify whether PCR and RCA were conducted on both inoculated and systemic tissues in each treatment group.
Ans: The PCR and RCA are carried out on both systematic and inoculated leaves to check the viral replication in the plant. And this information is updated in the revised version.
Q24- Line 348: Missing clear numeric results of the qPCR assays. Add the exact fold reduction in viral load from qPCR to strengthen your conclusion of attenuation. Although present on figure 6, stating it on the main text will help better understand.
Ans: Following explanation is added in manuscript
To evaluate the viral load in plants inoculated with ToLCNDV, quantitative PCR (qPCR) analysis was performed. Plants co-inoculated with CRISPR cassettes 1 and 2, along with ToLCNDV, exhibited significantly reduced viral titers (mean ±â€¯SD: 0.09 ±â€¯0.03 and 0.11 ±â€¯0.02, respectively) compared to the control plants inoculated with ToLCNDV alone (1.00 ±â€¯0.05). Plants treated with CRISPR cassettes 3 and 4 showed a moderate decrease in viral titers (0.37 ±â€¯0.04 and 0.35 ±â€¯0.05, respectively). Each treatment group included three biological and three technical replicates. Statistical analysis using one-way ANOVA confirmed a significant reduction in viral load for constructs 1 and 2 (P < 0.01). These results are consistent with the observed suppression of disease symptoms and demonstrate the effectiveness of CRISPR-Cas9 constructs in targeting ToLCNDV. Notably, CRISPR cassettes 1 and 2 led to approximately a 10-fold reduction in viral accumulation compared to the control, while constructs 3 and 4 achieved around a 3-fold decrease. These findings highlight the superior performance of constructs 1 and 2 in reducing ToLCNDV replication and underscore the potential of the gRNA-Cas9 approach as a targeted antiviral strategy.
Q25- Line 366: The main
Ans: The phrase ‘Main objective of this study was’ has been revised to ‘The goal of this study was”
Q26- Line 367: again nomenclature style problem, please revise it on the entire text.
Ans: We have carefully reviewed the entire manuscript and revised all virus name.
Q27- Line 369: change results to findings
Ans: Thank you for the suggestion. We have replaced the word “results” with “findings” to improve the clarity and tone of the sentence.
Q28- Line 370-371: down-regulates viral accumulation and symptoms 370 in Nicotiana benthamiana plants change to reduce viral accumulation and mitigates symptom expression in Nicotiana benthamiana
Ans: The phrase ‘down-regulates viral accumulation and symptoms in Nicotiana benthamiana plants’ has been revised to “effectively reduces viral accumulation and mitigates symptom expression in Nicotiana benthamiana.”
Q29- Line 371: remove conducted here
Ans: We have removed the word "conducted" from the sentence.
Q30- Line 372: for engineering resistance to plant viruses
Ans: Line 488, The phrase ‘genome-editing tools to develop resistance against plant viruses’ has been revised to ‘genome-editing tools for engineering resistance to plant viruses’.
Q31- Line 373: change “Major impact of ToLCNDV on economically important crops like “ to Given the substantial impact of ToLCNDV on economically important crops such as.
Ans: The sentence has been revised as “Given the substantial negative effects of ToLCNDV on economically important crops.”
Q32- Line 375-376: down-tone “making it essential to explore other approaches” to highlighting the necessity of exploring alternative methods
Ans: The phrase has been revised to “Because traditional breeding efforts have been not succeeded in developing resistant cultivars, alternative methods are needed.”
Q33- Line 376-377: Same problem “CRISPR-Cas9, known for its precision and adaptability [4, 19, 33], offers a promising path forward”
Ans: The phrase has been revised with “The CRISPR-Cas9 editing system [58] that brings editing precision and flexibility [59, 60] offers a promising path forward.”
Q34-Line 394-395: viral nomemclature – please fix it
Ans: We have revised the viral nomenclature in this line to follow the ICTV. The common virus name remains non-italicized as Tomato leaf curl New Delhi virus, in accordance with current ICTV guidelines.
Q35- Line 396: replace "prone" with susceptible
We have replaced the word “prone” with “susceptible” in the revised manuscript.
Q36- Can be add on the introduction due to their relevance: Zhan et al 2022, 2024 https://doi.org/10.3390/plants13233313;https://doi.org/10.3390/ijms23042303 These references offer significant insights into CRISPR-based strategies for plant-virus resistance and are good reviews for this topic.
Ans: These references were added in the introduction due to relevance in Line 90 after this statement, ‘CRISPR/Cas9-based genome editing is an effective tool for improving genetics and combating various diseases in plants [7, 14]‘
Once these revisions have been incorporated, I believe the manuscript will be much stronger for publication. I congratulate the authors on their innovative approach and the potential impact of their work on plant viral resistance.
We are thankful to the reviewer for positive criticism and encouraging statement. We hope to fulfill the requirements of manuscript as a better publication.
Reviewer 3 Report
Comments and Suggestions for Authors
In this manuscript, authors investigated the use of multiple guide RNAs (gRNAs) with CRISPR-Cas9 to target and reduce Tomato Leaf Curl New Delhi Virus (ToLCNDV) in plants. The authors designed and tested several gRNAs targeting conserved regions of the virus, selected the most effective ones, and combined them for a multiplexed approach. The constructs were tested in Nicotiana benthamiana plants, showing reduced virus symptoms and viral load. This is the first report to demonstrate the effectiveness of multiplexed gRNA against ToLCNDV. The idea is great and the design of experiments are fine. I have the following comments.
Major comments
The methods section lacks specific details about the transient expression system, such as the exact agroinfiltration protocol, controls used, number of replicates, and statistical analysis. It is always good to describe materials and methods in detail.
The off-target analysis for gRNA selection is mentioned but not fully described; experimental validation of off-target effects is missing.
The results of the in vitro cleavage assay are not shown quantitatively (gel images or efficiency percentages are absent).
The manuscript does not provide detailed quantitative data (error bars and statistical significance) for viral load reduction in plants.
The discussion on the potential for viral escape and the practical challenges of moving from transient assays to stable crop resistance is limited.
Minor comments
Some tables and figures are dense or not clearly formatted; legends and references to them in the text could be improved.
There are minor typographical and grammatical errors; the text would benefit from careful proofreading and simpler sentence structure.
Acronyms are not always defined at first use.
The reference list and in-text citations should be checked for completeness and formatting.
Supplementary data (raw qPCR data and sequence alignments) could be referenced if available.
Virus names should be written according to ICTV rules. Check them thoroughly.
In my opinion, this study addressed an important problem and uses a promising and challenging approach; however, this manuscript would be stronger with more methodological detail, quantitative data, and expanded discussion. Therefore, I recommend major revisions.
Comments on the Quality of English LanguageThe English could be improved to more clearly express the research.
Author Response
Reviewer 3:
Comments and Suggestions for Authors
In this manuscript, authors investigated the use of multiple guide RNAs (gRNAs) with CRISPR-Cas9 to target and reduce Tomato Leaf Curl New Delhi Virus (ToLCNDV) in plants. The authors designed and tested several gRNAs targeting conserved regions of the virus, selected the most effective ones, and combined them for a multiplexed approach. The constructs were tested in Nicotiana benthamiana plants, showing reduced virus symptoms and viral load. This is the first report to demonstrate the effectiveness of multiplexed gRNA against ToLCNDV. The idea is great and the design of experiments are fine. I have the following comments.
Q1- The methods section lacks specific details about the transient expression system, such as the exact agroinfiltration protocol, controls used, number of replicates, and statistical analysis. It is always good to describe materials and methods in detail.
Ans: We have updated the section 2.5. Plants inoculation. Where we added the necessary details about agro-infilteration protocols.
Q2- The off-target analysis for gRNA selection is mentioned but not fully described; experimental validation of off-target effects is missing.
The computational off-target effects were analyzed using CRISPRdirect and a BLAST search from NCBI. The details have been revised in the manuscript as follows:
“The next-tier of gRNA selection was based on the results of off-target analysis using CRISPRdirect [45] and BLAST searches from the NCBI to detect that either the chosen gRNA sequence match or target the chili genome or Nicotiana benthamiana genome region. Additionally, analysis was carried out to detect single nucleotide polymorphism (SNPs) in ToLCNDV genome sequences to ensure that the predicted target regions were conserved among all ToLCNDV isolates. “
Q3- The results of the in vitro cleavage assay are not shown quantitatively (gel images or efficiency percentages are absent).
We thank the reviewer for this observation. In response, we have now included a paragraph in the section 2.3.
Q4- The manuscript does not provide detailed quantitative data (error bars and statistical significance) for viral load reduction in plants.
Reviewer 2 also pointed out this issue. We have already added the quantitative data to the manuscript, as shown below.
‘To evaluate the viral load of plants inoculated with ToLCNDV using qPCR, plants inoculated with CRISPR cassette 1 and 2, coupled with ToLCNDV, showed significantly lower viral titers (mean±SD: 0.09± 0.03 and 0.11± 0.02, respectively) compared to control plants inoculated with ToLCNDV (1.00± 0.05). Plants treated with CRISPR cassette 3 and 4 showed a moderate decline in viral titers (0.37± 0.04 and 0.35± 0.05, respectively). Each group consisted of three biological and three technical replicates. We used one-way ANOVA for statistical analysis, which showed a significant reduction in viral load for constructs 1 and 2 (P < 0.01). These results correspond to the observed symptoms of virus suppression, demonstrating the efficiency of the constructed CRISPR cassette in targeting ToLCNDVThese results correlate with the observed suppression of disease symptoms, supporting the efficacy of the gRNA-Cas9 approach in targeting ToLCNDV.
Plants transformed with CRISPR cassettes 1 and 2 showed low viral accumulation (0.09 and 0.11) compared to the control (1.00), while constructs 3 and 4 showed values of 0.37 and 0.35, respectively. Although these also represent significant reductions, constructs 1 and 2 demonstrated more promising results, exhibiting a 10-fold decrease in viral accumulation compared to the control, whereas constructs 3 and 4 showed a 3-fold reduction, as supported by ANOVA.’
Q5- The discussion on the potential for viral escape and the practical challenges of moving from transient assays to stable crop resistance is limited.
The discussion section is updated and we provided the information regarding practical challenges mentioned by the reviewer.
Minor comments
Q6- Some tables and figures are dense or not clearly formatted; legends and references to them in the text could be improved.
We thank the reviewer for this helpful observation. In the revised manuscript, we have carefully reviewed all tables and figures for clarity and formatting. Legends have been rewritten to provide clearer, more detailed explanations of the data presented. We have also improved the referencing of figures and tables within the main text to ensure better integration and flow. These revisions enhance the readability and interpretability of the results.
Q7- There are minor typographical and grammatical errors; the text would benefit from careful proofreading and simpler sentence structure.
We have carefully proofread the manuscript to correct the minor typographical and grammatical errors. Additionally, we have revised the text to simplify sentence structures where necessary.
Q8- Acronyms are not always defined at first use.
We have carefully reviewed the manuscript to ensure that all acronyms are defined at their first appearance in the text.
Q9-The reference list and in-text citations should be checked for completeness and formatting.
We have carefully reviewed the reference list and all in-text citations to ensure completeness, accuracy, and proper formatting according to the journal's guidelines.
Q10- Supplementary data (raw qPCR data and sequence alignments) could be referenced if available.
The sequence alignment data is very simple and it is difficult to put in one section or figure. The window of 2750 nucleotides alignment for 346 sequences makes it difficult to view until opened in the respective software. However, we can provide the fasta file for the sequences we used for alignment.
Q11- Virus names should be written according to ICTV rules. Check them thoroughly.
We have thoroughly reviewed all virus names throughout the manuscript and revised them according to ICTV rules.
Q12-Comments on the Quality of English Language. The English could be improved to more clearly express the research.
We have carefully revised the manuscript to improve clarity, grammar, and overall readability. The text has been edited to ensure it meets the standards expected in scientific writing and more effectively communicates the research. We are confident that the revised version now clearly presents our findings.
In my opinion, this study addressed an important problem and uses a promising and challenging approach; however, this manuscript would be stronger with more methodological detail, quantitative data, and expanded discussion. Therefore, I recommend major revisions.
Round 2
Reviewer 1 Report
Comments and Suggestions for Authors
Compared to the original version of the manuscript, the paper was considerably improved. However, there are still concerns that must be addressed before the paper can be accepted for publication.
- The authors did not perform the additional control experiments requested in the first round of reviewing, which would prove that using two gRNAs is more efficient than using a single gRNA in the experimental system described. Therefore, there is no reason to make any statements about the efficiency of the methodology used. Thus, any statements about the efficiency of the approach involving two gRNAs should be removed, including the title of the paper.
- Since the authors did not perform the requested additional experiments, the amount of experimental work reported remained insufficient for a full-size paper. Therefore, the manuscript should be converted to the "Brief Report" format. All technical and less important information, including tables and some gel pictures, should be moved to the supplementary materials.
Author Response
Answers to reviewer
- Compared to the original version of the manuscript, the paper was considerably improved. However, there are still concerns that must be addressed before the paper can be accepted for publication.
Ans. We are thankful to the reviewer for his valuable suggestions. Here we have provided further details for better presentation of the manuscript.
Q-1. The authors did not perform the additional control experiments requested in the first round of reviewing, which would prove that using two gRNAs is more efficient than using a single gRNA in the experimental system described. Therefore, there is no reason to make any statements about the efficiency of the methodology used. Thus, any statements about the efficiency of the approach involving two gRNAs should be removed, including the title of the paper.
Ans-1: We appreciate the reviewer’s concerns regarding the additional control experiments. In our manuscript, we have cited multiple previous studies that demonstrate resistance developed using single guide RNA (gRNA) is often not durable due to the emergence of escape mutants. Our constructs 3 and 4 target single viral regions (CP and Rep, respectively) and serve as analogs to these previously published results. Constructs 1 and 2, in contrast, target multiple regions simultaneously using multiplexed gRNAs.
Our data show that constructs 1 and 2, which target two distinct viral regions, result in reduced viral accumulation compared to constructs 3 and 4, supporting the concept that targeting multiple regions enhances resistance efficacy. We believe that repeating the well-established single-gRNA experiments would be redundant and not add new insights beyond existing literature.
Nonetheless, to avoid any confusion or overstatement regarding the efficiency of the multiplexed approach, we have revised the manuscript title and tempered claims about efficiency, in agreement with the reviewer’s suggestion. We trust this addresses the concerns and clarifies our experimental rationale. Now the new title of the manuscript is, “Potential for duplexed, in-tandem gRNA-mediated suppression of two essential genes of Tomato leaf curl New Delhi virus in crop plants”.
Q-2. Since the authors did not perform the requested additional experiments, the amount of experimental work reported remained insufficient for a full-size paper. Therefore, the manuscript should be converted to the "Brief Report" format. All technical and less important information, including tables and some gel pictures, should be moved to the supplementary materials.
Ans-2. We appreciate the reviewer’s suggestion regarding the manuscript format and the presentation of experimental data. However, we believe that the amount of experimental work presented provides a comprehensive comparative analysis of targeting single versus multiple viral regions using CRISPR-Cas9. The included gel images and related data are integral to supporting the key findings and validating the experimental outcomes.
Removing these results or relegating them to supplementary materials could reduce the clarity and impact of the manuscript. Therefore, we respectfully disagree with converting the manuscript to a “Brief Report” format or moving essential figures and tables to supplementary files. We believe the current format appropriately reflects the scope and significance of the study.
Reviewer 3 Report
Comments and Suggestions for Authors
The authors have thoroughly addressed the reviewer’s comments and made the necessary revisions.
Author Response
The reviewer has appreciated our manuscript's revisions. We are thankful to reviewer for being positive and encouraging the publication of our findings.
Round 3
Reviewer 1 Report
Comments and Suggestions for Authors
The authors changed the wording as requested, but they decided not to convert the paper into a brief report, which, in my view, would be appropriate.
Thus, I believe the paper can be accepted for publication in its current form, provided the editor believes the amount of reported data is sufficient for a full-size paper.